# Detection of Carrageenan in Cheese Using Lectin Histochemistry

**Marie Bartlová** , **Matej Pospiech** , **Zdeňka Javůrková \*** and **Bohuslava Tremlová**

Department of Plant Origin Food Sciences, Faculty of Veterinary Hygiene and Ecology, University of Veterinary Sciences Brno, 612 42 Brno, Czech Republic; bartlovam@vfu.cz (M.B.); mpospiech@vfu.cz (M.P.); tremlovab@vfu.cz (B.T.)
\* Correspondence: javurkovaz@vfu.cz; Tel.: +420-541-562-704

**Abstract:** Carrageenan is a substance widely used as an additive in the food industry. Among other things, it is often added to processed cheese, where it has a positive effect on texture. Processing of such cheese involves grinding, melting and emulsifying the cheese. There is currently no official method by which carrageenan can be detected in foodstuffs, but there are several studies describing its negative health impact on consumers. Lectin histochemistry is a method that is used mainly in medical fields, but it has great potential to be used in food analysis as well. It has been demonstrated that lectin histochemistry can be used to detect carrageenan in processed cheese by Human Inspection and Computer-Assisted Analysis (CIE L\*a\*b\*). The limit of detection (LoD) was established at 100 mg kg$^{-1}$ for Human Inspection and 43.64 for CIE L\*a\*b\*. The CIE L\*a\*b\* results indicate that Computer-Assisted Analysis may be an appropriate alternative to Human Inspection. The most suitable parameter for Computer-Assisted Analysis was the b\* parameter in the CIE L\*a\*b\* color space.

**Keywords:** agglutinins; *Arachis hypogaea*; CIE L\*a\*b\*; fixation; hydrocolloids; food; light microscopy

## 1. Introduction

Carrageenan is a polysaccharide that consists of D–galactose and 3,6–anhydro–galactose units that are linked by α–1,3 and β–1,4 glycosidic bonds. These polysaccharides are obtained from red seaweed (Rhodophyceae) [1–3]. Carrageenan is used in the food industry mainly for its gel-forming, thickening, emulsifying, and stabilizing properties in the amount of 0.005–2.0%. The K-, ι- and λ-carrageenans are of the greatest commercial importance. They are used, for example, to improve the texture of curd cheese, blancmange, dairy desserts, cheese, ice cream, and meat products [4–7]. The use of carrageenan in foods is considered safe. However, there are studies indicating that carrageenan can cause inflammatory reactions, damage the digestive mucosa, or be associated with breast cancer [3,8].

Processed cheese is a good source of protein, fat, minerals, and vitamins. The basic raw materials are natural cheese, butter, emulsifying salts, and water [9]. The addition of carrageenan prevents syneresis of dairy products and it has thickening and gelling properties [10]. The strength of the gel depends on the carrageenan–casein binding in cheese with the addition of carrageenan. In particular, this bond is formed between the negatively charged sulphate groups contained in the carrageenan and the positively charged region of the casein. This binding depends on parameters such as hydrocolloid and protein concentration, sugar content, temperature, or pH. Carrageenan stabilizes the fat in cheese and may even serve as a substitute for the emulsifying salt, which may lead to a reduction in the amount of phosphorus and sodium in processed cheese [9–11].

The principle of lectin histochemistry lies in the lectin–saccharide bond. Lectins are proteins and glycoproteins that bind to specific carbohydrates [12]. Lectin histochemistry uses this principle, by which glycan residues can be detected directly or indirectly. Direct

methods are based on the binding of lectin that is conjugated to a particular carbohydrate by a fluorochrome or enzyme. Indirect methods involve the binding of a labeled antibody to lectin that binds to a given carbohydrate. In addition, methods based on the avidin–biotin bond can be used, in which biotinylated lectin binds to the saccharide. In the next step, the avidin–biotin complex, which is conjugated to a fluorochrome or enzyme, is attached to it [12–14].

Carrageenan is an additive that can be used to adulterate foods. For this reason, it is necessary to be able to detect them. There is currently no official method for their detection in food. In this respect, the use of lectin histochemistry in food analysis, which has not yet been used in this field, offers great potential. The aim of this paper is to validate the detection of carrageenan in cheese using lectin histochemistry and comparing the results of Human Inspection measurements and CIE L*a*b* measurements.

## 2. Materials and Methods

This work took place in several steps, which are shown in Figure 1. First, the method of sample fixation and the concentration of lectins used to test individual cheese samples were tested. Subsequently, the parameters of CIE L*a*b* and LoD were determined.

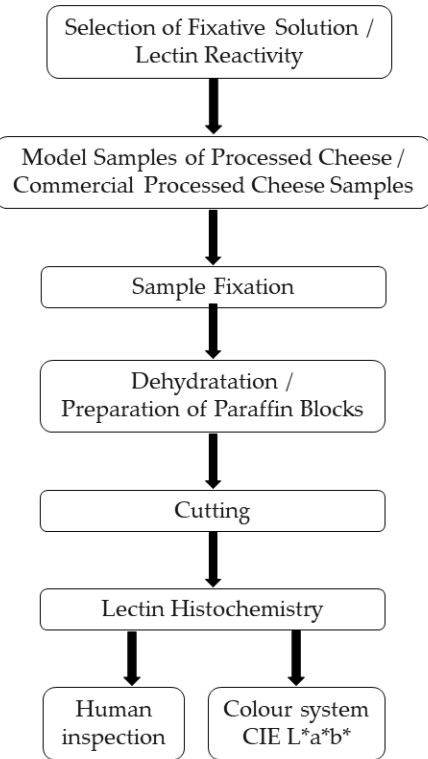

**Figure 1.** Methodology and analysis scheme.

### 2.1. Choice of Fixative Solution

In the first phase of testing, two samples of processed cheese were purchased from different manufacturers in the retail market, and they were used to test and select the most suitable method of processed cheese fixation. The samples were cut into 1 cm$^3$ parts and fixed with selected fixative solutions that are used in histology most often. These solutions included the Alcohol Formalin Acetic Acid (AFA) solution prepared according to Sánchez-Pardo et al. [15], a solution of 10% formalin (Penta, Prague, Czech Republic), 100% ethanol, and Carnoy's solution prepared according to Saulacic et al. [16]. A sample that was not fixed was used as the control to check the results of changes in fixation. Fixation lasted for 48 h. The samples were then measured for firmness on a TA.XTplus Texture Analyzer (Stable Micro Systems, Godalming, United Kingdom) at room temperature. Samples of

cheese were centrally placed on the heavy-duty platform and a 5 mm puncture probe (SMS P/5) was used for the penetration test around the mid-region of the cheese. The probe penetrated the sample of cheese to 2 mm, and the test speed was adjusted at 1 mm·s$^{-1}$.

The results were statistically evaluated by multiple comparisons with t-distribution of Kruskal–Wallis One-Way ANOVA.

### 2.2. Production of Model Samples of Processed Cheese

Subsequently, model samples of processed cheese were made from 250 g of ripened hard cheese (30% Eidam, Lidl, Neckarsulm, Germany). The cheese was ground, 13 g of butter (Madeta, České Budějovice, Czech Republic), 200 mL of water, 13 g of emulsifying salts (Fosfa, Břeclav, Czech Republic), and carrageenan were added. The carrageenan used included: κ-carrageenan (Sigma-Aldrich, St. Louis, MO, USA), ι- and λ-carrageenan (Eurogum, Herlev, Denmark) at concentrations of 100 mg kg$^{-1}$, 1000 mg kg$^{-1}$, and 10,000 mg kg$^{-1}$. The mixture was melted at 90 °C for 10 min in a Vortex Thermomix (Vorwerk, Wuppertal, Germany). A control sample without carrageenan was also prepared.

In addition, 27 samples of processed cheese were purchased from the retail market. Samples were selected at random. Twelve samples had no carrageenan as their stated ingredient, while 15 samples showed carrageenan in their ingredients list.

Processed cheese samples were cut into 1 cm$^3$ portions. Based on the results from the first part of the study, the samples were fixed with Alcohol Formalin Acetic Acid solution for at least 24 h in the next step. Afterwards, the samples were dehydrated by an ascending alcohol series in an autotechnicon (TP 1020, Leica, Wetzlar, Germany). Subsequently, the cheese samples were embedded in paraffin (Leica-paraplast plus, Leica Microsystems Vertrieb, Wetzlar, Germany). From each processed cheese sample, 4 paraffin blocks were prepared and cut on a rotating microtome (RM2255, Leica, Wetzlar, Germany) into 5 μm thick sections on SuperFrost® Plus glass (Thermo Scientific, Waltham, MA, USA). Four sections were cut from each block, which were then dried in a thermostat (Memmert, Büchenbach, Germany).

### 2.3. Lectin Histochemistry

Lectin histochemistry is performed based on the methods by Bartlová et al. [17]. Biotinylated *Arachis hypogaea* lectin (peanut agglutinin, PNA) (Sigma-Aldrich, St. Louis, MO, USA) at a concentration of 2 μg mL$^{-1}$ was used to detect carrageenan. Lectin of *Arachis hypogaea* binds specifically to sugars such as galactose and N-acetylgalactosamine by non-covalent bonds such as van der Walls and hydrogen bonds [12,17–19].

B-Calleja solution was used for background staining which is prepared by mixing 1.0 mL of distilled water, 1.0 g of indigo carmine, and 200 mL of picric acid. Afterwards the solution was filtered [17].

Signal Intensity Evaluation

- Human Inspection

The samples were evaluated by visual scoring of DAB by precipitation using light microscope of Eclipse Ci-L (Nikon, Minato, Japan) with medium to strong magnification (20× and 40×). A sample with 5 or more positive sections was evaluated as a positive sample. In this work, Human Inspection was considered to be the reference method.

- Computer-Assisted Analysis of CIE L*a*b*

Scanning of stained model cheese samples and samples purchased in the retail market was performed with a DFK 23U274 camera (Imaging Source, Bremen, Germany) using an Eclipse Ci-L microscope (Nikon, Minato, Japan) with a Prosca III motorized stage (Prior, Rockland, MA, USA). NIS-Elements AR 5.20 software (Laboratory Imaging, Praha, Czech Republic) was used to scan the samples. The L*a*b* parameters were measured on 80 randomly selected fields of view (4 blocks, 2 sections for each sample), where L* is the lightness (0–100), a* indicates the position on red-green axis (+a to −a) and b* on the

yellow–blue axis (+b to −b). Spectral analysis was performed using a USB4000-UV-VIS-ES microspectrophotometer (Ocean Optics Inc., Orlando, FL, USA) with a 0.025 mm$^2$ probe.

### 2.4. Statistical Processing

The data were processed statistically using the 2014.5.03 XLSTAT software (Addinsoft, Paris, France), while the statistical significance of the obtained results was determined at a significance level of $\alpha$ = 0.05. The normality test confirmed the normal distribution of the data. An ANOVA Tukey HSD test was used to compare L*, a* and b*. McNemar's test was used to compare Human Inspection and CIE b*. Individual value plot show variability of each measurement in CIE b*, where X = b* and Y = samples. Between each group there is a 50-point distance for better data visualization

## 3. Results and Discussion

### 3.1. Choice of Fixative Solution

Fixation is one of the key steps in the preparation of samples for histological and immunohistochemical methods, and it has a significant effect on the quality of the whole method and the results obtained. Samples should be prepared to preserve their structure [20]. Fixation prevents tissue decomposition and also minimizes damage during other processes, such as dehydration, watering, and sample cutting [21]. It has been found that formaldehyde fixation and histological processing can lead to denaturation and epitope masking [22]. Immunoreactivity may be directly proportional to the storage of slides [23]. For some matrices, sample processing is already a routine practice. In particular, the histology of animal tissues [24,25] and plant tissues [25–27]. Several studies are also available for foodstuffs of meat origin describing the method of their fixation [28,29]. Processing for microscopic methods has also been described for cheese [9,30,31]. During fixation, the sample is intensively cross-linked and its structure is stabilized [32–34]; therefore, the hardness parameter of the sample was chosen as an indicator of suitable fixation. Extensive cross-linking can lead to loss or reduction of enzymatic activities [35]. The differences in firmness between the individual samples are shown in Figure 2.

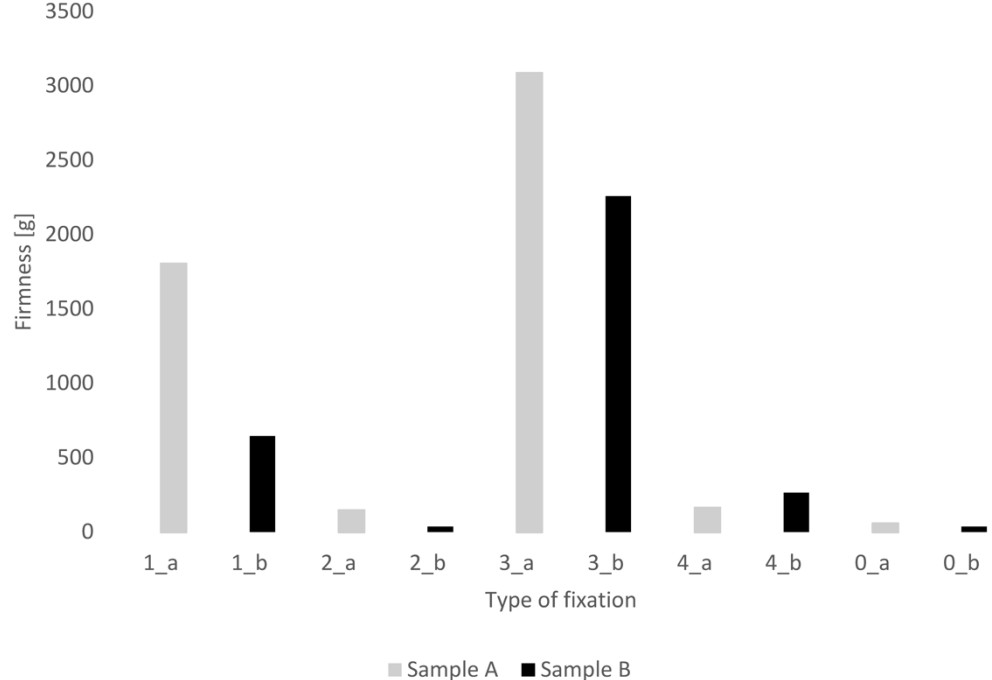

**Figure 2.** Comparison of texture measurement (firmness) results for processed cheese samples [g]. Effect of fixation to the firmness of processed cheese samples. Notes: 0—no fixation, 1—AFA, 2—10% formol, 3—100% ethanol, 4—Carnoy's solution.

When comparing the texture of the fixed samples with the control sample that was not fixed (0_a—58.47 g and 0_b—37.53 g), the samples fixed in Carnoy's solution (4_a —164.43 g and 4_b—266.03 g) and in 10% formol (2_a—147.92 g and 2_b—36.98 g) had a similar consistency. The consistency of these samples was very soft and non-compact. Formalin can cause tissue swelling and can give cells a vitreous appearance [36]. In contrast, the highest texture values were recorded for samples that were fixed with ethanol (3_a—3087.34 g and 3_b—2257.79 g). The consistency of these samples was friable. The reason for this result is probably the dissolved lipids. Alcohol-based fixing solutions penetrate tissue poorly, precipitate proteins and dissolve lipids. Long-term fixation in absolute alcohol may lead to tissue shrinkage [37,38]. High hardness and fragility can cause tissue damage during following sample manipulations [21]. Carnoy's solution also contains alcohol. However, the chloroform and acetic acid, which are in this solution too, prevent the shrinking of the tissue caused by ethanol [32]. The results show that the fixations (10% formol, ethanol and Carnoy's solution) are not suitable for processing of processed cheese samples.

The graph shows that samples fixed in AFA fixative solution (1_a—1805.38 g and 1_b—645.65 g) have the most suitable consistency because these samples were not soft and remained compact. A suitable fixation has to strengthen and make it hard enough to minimize the damage and reduce the deformation during dehydration, embedding, and cutting [21]. The differences in values between the samples are most likely given by composition. For example, hydrocolloids have a significant impact on the texture profile, such as hardness, cohesion, springiness, chewing, and gumminess [39].

A highly significant difference ($p < 0.01$) was found between the individual samples of fixative solutions in comparison with the control sample except for the sample fixed in 10% formalin ($p > 0.05$).

### 3.2. Lectin Histochemistry

3.2.1. Lectin Concentration

Prior to the analysis itself, the correct lectin concentration to be used in the analysis must be determined. Brooks et al. [12] recommends first diluting lectin to a concentration of 10 µg mL$^{-1}$, because this concentration is optimal for most lectins in indirect methods of lectin histochemistry and then adjust the concentration for optimal results. In the case of high concentrations of lectins, a false positive reaction can occur [40].

The *Arachis hypogaea* lectin concentrations of 10 µg mL$^{-1}$, 2 µg mL$^{-1}$, and 1 µg mL$^{-1}$ were tested on a sample of processed cheese from the retail market (Table 1) in this work. These concentrations were selected following previous work in which carrageenan was determined in meat products [17].

**Table 1.** Validation of *Arachis hypogaea* lectin reactivity on a sample of processed cheese.

| Lectin Concentration | Color Intensity | L* | a* | b* |
|---|---|---|---|---|
| 10 µg mL$^{-1}$ | +++ | 81.54 ± 9.18 [a] | −2.63 ± 2.01 [a] | 18.17 ± 5.73 [b] |
| 2 µg mL$^{-1}$ | +++ | 86.05 ± 5.68 [ab] | −5.08 ± 0.94 [b] | 17.91 ± 4.22 [a] |
| 1 µg mL$^{-1}$ | +++ | 89.82 ± 3.73 [b] | −6.02 ± 0.94 [b] | 11.77 ± 3.53 [a] |

Note: Mean values denoted by different letters at individual column are significantly different ($p < 0.05$). Data are expressed as mean value ± standard deviation. Signal intensity is from + (weak) to +++ (very strong).

For the L* and a* parameters, a statistically significant difference was confirmed for the tested concentrations of 1 and 10 µg mL$^{-1}$ ($p < 0.05$). No statistical difference was demonstrated for the b* parameter. The L* parameter confirms that the higher the lectin concentration, the higher the signal intensity (darker histological tissue staining). The lectin concentration of 2 µg mL$^{-1}$ was selected as optimal, where the color intensity is comparable to the highest lectin concentration of 10 µg mL$^{-1}$, while a lower concentration allows the cost of a single analysis to be reduced. This concentration was also used for the following analyses.

### 3.2.2. Limit of Detection

The limit of detection (LoD) is an important parameter of all analytical methods [41]. Histological methods most often have a qualitative expression. Therefor qualitative detection methods are also selected. In the case of LoD, it is an expression of the number of positive findings in the examined sections. For the validated lectin histochemistry, the LoD was determined for all three types of carrageenan per 100 mg kg$^{-1}$ (Table 2).

**Table 2.** Limit of detection for Human Inspection.

| Sample | Carrageenan Type | Carrageenan Concentration [mg kg$^{-1}$] | Number of Positive/Negative Sections * | Correct Rate |
|---|---|---|---|---|
| LD1A | κ-carrageenan | 100 | 6/2 | 75% |
| LD1B | κ-carrageenan | 1000 | 8/0 | 100% |
| LD1C | κ-carrageenan | 10,000 | 8/0 | 100% |
| LD2A | ι-carrageenan | 100 | 8/0 | 100% |
| LD2B | ι-carrageenan | 1000 | 7/1 | 87.5% |
| LD2C | ι-carrageenan | 10,000 | 6/2 | 75% |
| LD3A | λ-carrageenan | 100 | 8/0 | 100% |
| LD3B | λ-carrageenan | 1000 | 3/5 | 60% |
| LD3C | λ-carrageenan | 10,000 | 8/0 | 100% |
| LD40 | Control sample (free from carrageenan) | 0 | 0/8 | 100% |

* a positive sample means 5 or more positive sections (62.5%).

With the development of computing technologies and digital microscopy, it is now possible to express the result of histological examination also by quantitative methods [42], which use image analysis or one of the other microscopic techniques that display data digitally, not only visually [43]. For quantitative expression in histological techniques, it is necessary that the test substance is specifically bound and highlighted by a specific substance. In our study, carrageenan is selectively labeled with lectin histochemistry and visualized by 3,3′-diaminobenzidine (DAB). The possibility of demonstrating the intensity of the DAB-bound signal was verified using the spectroscopic method measured in the CIE L*a*b* color system. According to one study [44], a specific component in the CIE L*a*b* system is the b* component, which includes values ranging from blue to yellow, ranging from approximately −128 to 128 [45,46]. This statement was also confirmed by our study, where with increasing carrageenan concentration, and thus higher intensity of DAB signal, the value of b* decreased. In the CIE L*a*b* system, this means a darker color (Table 3).

**Table 3.** Limit of detection for CIE L*a*b* system.

| Sample | Carrageenan Type | Carrageenan Concentration [mg kg$^{-1}$] | L* | a* | b* |
|---|---|---|---|---|---|
| LD1A | κ | 100 | 81.31 ± 6.06 [de] | −12.66 ± 4.82 [a] | 38.12 ± 7.82 [bc] |
| LD1B | κ | 1000 | 86.62 ± 4.58 [bc] | −12.89 ± 2.33 [a] | 35.18 ± 5.76 [cd] |
| LD1C | κ | 10,000 | 88.28 ± 4.20 [ab] | −18.44 ± 1.59 [f] | 31.80 ± 6.18 [ef] |
| LD2A | ι | 100 | 26.2 ± 3.67 [c] | −13.30 ± 1.16 [ab] | 40.52 ± 4.15 [b] |
| LD2B | ι | 1000 | 81.51 ± 4.62 [d] | −15.89 ± 2.63 [de] | 36.63 ± 6.55 [cd] |
| LD2C | ι | 10,000 | 86.60 ± 3.77 [bc] | −16.59 ± 3.72 [e] | 35.42 ± 6.33 [cd] |
| LD3A | λ | 100 | 89.71 ± 2.57 [a] | −14.49 ± 1.19 [bc] | 36.32 ± 5.64 [cd] |
| LD3B | λ | 1000 | 76.57 ± 3.18 [f] | −14.88 ± 1.86 [cd] | 34.16 ± 4.88 [de] |
| LD3C | λ | 10,000 | 87.99 ± 4.64 [ab] | −13.83 ± 2.08 [abc] | 29.48 ± 5.55 [f] |
| LD40 | Control | 0 | 78.95 ± 5.76 [e] | −14.15 ± 3.67 [abc] | 43.64 ± 4.11 [a] |

Note: Mean values denoted by different letters at individual column are significantly different ($p < 0.05$). Data are expressed as mean value ± standard deviation.

The results also confirmed a statistically significant difference between the control and the samples containing carrageenan additions ($p < 0.05$) for b* parameter. L* and a* did not differ statistically for some low addition concentrations or for 1% λ-carrageenan addition. The reason for this is the lower colorability of λ-carrageenan.

For quantitative methods, LoD is expressed as 3 times the noise. For quantitative lectin histochemistry, LoD was determined to be $43.64 \pm 12.33$ of b* value.

### 3.2.3. Validation of the Method on Samples from the Retail Market

Detecting carrageenan in food is very problematic. Thus, there is no general method for its analysis yet [17,47]. Several methods for detecting carrageenan in food have been described in the literature. These include, for example, colorimetric methods [17,48,49]. Soedjak [50] detected carrageenan in milk photometrically with the addition of methylene blue. Ziółkowska et al. [51] used photometric titration to detect carrageenan. Quantitative determination of carrageenan in infant formula, chocolate milk, and ice cream has been performed using a method based on the degradation of proteins by papain and subsequent precipitation of carrageenan with Hyamine solution and addition of phenol-sulfuric acid to produce a color change in carrageenan [52,53]. Other detection methods have used included potentiometry [54], chromatography [47] or microscopic methods [17,55].

In our work, carrageenan was detected in samples of processed cheese purchased in the retail market. The samples were examined by Human Inspection and CIE b *. Human inspection was chosen as the reference method for this comparison. The results are presented in Table 4.

**Table 4.** Results comparison of the retail market samples by Human Inspection and by b* of CIE L*a*b*.

| Sample No. | Result | Declared Ingredient | Human Inspection LH | CIE b* | |
|---|---|---|---|---|---|
| | | | | Threshold Comparison | ANOVA Comparison |
| Control | 0/2 0/2 0/2 0/2 | N | N | N | $43.64 \pm 4.10$ [a] |
| 65-19 | 1/1 0/2 1/1 0/2 | N | N | N | $41.18 \pm 9.74$ [ab] |
| 66-19 | 0/2 0/2 0/2 0/2 | N | N | P | $17.73 \pm 2.63$ [hij] |
| 67-19 | 2/0 2/0 2/0 2/0 | N | P | P * | $25.54 \pm 8.63$ [de] |
| 68-19 | 2/0 2/0 2/0 2/0 | N | P | N | $35.61 \pm 4.45$ [c] |
| 69-19 | 0/2 0/2 0/2 0/2 | N | N | N | $39.39 \pm 4.30$ [b] |
| 70-19 | 2/0 2/0 2/0 2/0 | N | P | P | $22.89 \pm 5.64$ [ef] |
| 71-19 | 2/0 2/0 2/0 2/0 | N | P | P | $17.57 \pm 4.62$ [hij] |
| 72-19 | 2/0 2/0 2/0 2/0 | P | P | P * | $23.11 \pm 7.74$ [ef] |
| 73-19 | 2/0 2/0 2/0 2/0 | P | P | P | $19.33 \pm 4.73$ [ghi] |
| 74-19 | 2/0 2/0 2/0 2/0 | P | P | P | $20.95 \pm 4.97$ [fg] |
| 75-19 | 2/0 0/2 2/0 2/0 | P | P | P | $15.89 \pm 2.94$ [jk] |
| 76-19 | 2/0 2/0 2/0 2/0 | P | P | P | $20.64 \pm 5.63$ [fgh] |
| 77-19 | 2/0 2/0 2/0 2/0 | P | P | P | $13.87 \pm 2.47$ [k] |
| 78-19 | 2/0 2/0 2/0 2/0 | P | P | P | $20.50 \pm 6.30$ [fgh] |
| 79-19 | 2/0 2/0 2/0 2/0 | P | P | P | $15.54 \pm 2.88$ [jk] |
| 80-19 | 1/1 2/0 1/1 1/1 | P | P | P | $20.01 \pm 7.03$ [fgh] |
| 81-19 | 2/0 2/0 2/0 2/0 | P | P | P | $17.78 \pm 5.50$ [hij] |
| 82-19 | 2/0 1/1 2/0 0/2 | P | P | P | $16.05 \pm 3.12$ [jk] |
| 83-19 | 1/1 0/2 2/0 2/0 | P | P | P | $15.12 \pm 2.10$ [jk] |
| 146-19 | 2/0 2/0 2/0 2/0 | N | P | P | $15.13 \pm 5.19$ [jk] |
| 147-19 | 2/0 2/0 2/0 2/0 | N | P | P | $13.84 \pm 3.69$ [k] |
| 148-19 | 2/0 2/0 2/0 2/0 | N | P | P | $20.36 \pm 4.25$ [fgh] |
| 149-19 | 2/0 2/0 2/0 2/0 | P | P | P | $20.96 \pm 4.99$ [fg] |
| 151-19 | 2/0 2/0 2/0 2/0 | P | P | P | $18.14 \pm 5.91$ [ghij] |
| 152-19 | 2/0 2/0 2/0 2/0 | P | P | P | $13.92 \pm 2.79$ [k] |
| 153-19 | 2/0 1/1 2/0 2/0 | N | P | P * | $27.13 \pm 7.92$ [d] |
| 154-19 | 2/0 2/0 2/0 2/0 | N | P | P | $16.53 \pm 3.26$ [ijk] |

Note: number of positive sections/negative sections; * after adding the required measurement uncertainty (3 × SD), a positive result could not be guaranteed. Mean values denoted by different letters at individual column are significantly different ($p < 0.05$). Data are expressed as mean value ± standard deviation.

The results show that the declaration for most samples was identical to the examination of lectin histochemistry. There were no statistical differences demonstrated between the Human Inspection and CIE b* methods by McNemar's test ($p > 0.05$). On the contrary, the samples (67-19, 68-19, 70-19, 71-19, 146-19, 147-19, 148-19, 153-19, 154-19) that were declared not to contain carrageenan by the producer returned positive results according to lectin histochemistry. The reason for this may also be the non-inclusion of carrageenan in the ingredients list of the product. For example, carrageenan is one of the most commonly used substances for adulteration in meat and meat products [56]. Another reason is product contamination. These theories are supported by sample 154-19, which is shown in Figure 3. The figure contains brown fragments of possible carrageenan, although this sample should have been negative according to the manufacturer.

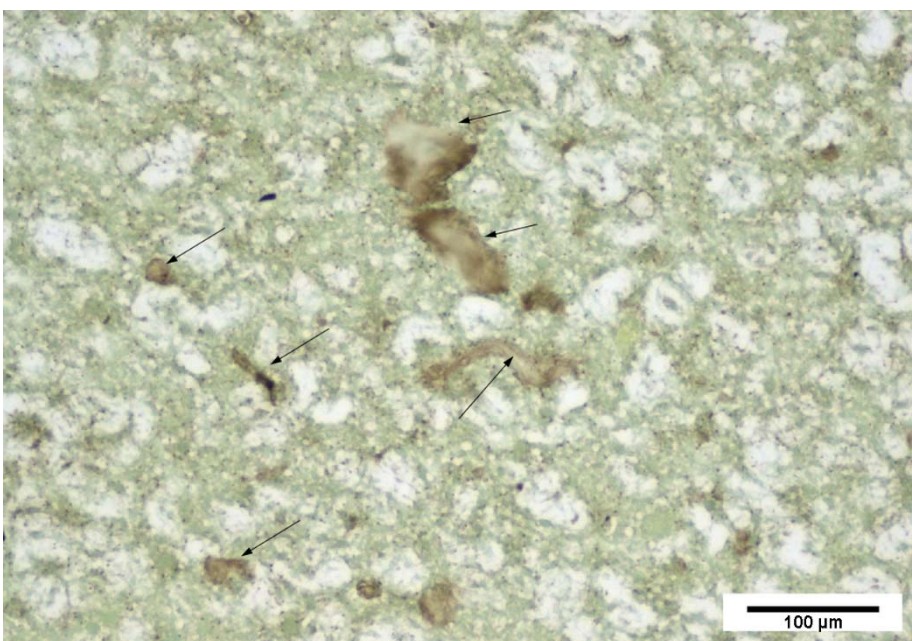

**Figure 3.** Positive sample of processed cheese, lectin histochemistry, background staining B-Calleja (green—protein, brown—carrageenan (arrows)).

Products with undeclared addition of carrageenan were examined by the CIE L*a*b* method as well. With regard to the quantitative result, there were not only differences between the positive and negative groups of samples, but also differences within these groups (Table 4). The reason for this is the different raw materials used as well as the different forms of carrageenan used. For clarity and interpretation of the variability of individual measurements, the results are also shown in scatter plots (Figure 4). The results confirm that the samples declared as positive (right) differed in the value of b* from the negative samples and, primarily, the negative control (left). Sample 66-19 differed from the negative samples with respect to b*, and was evaluated as negative by a human evaluator. However, according to the set threshold value for b*, this sample was evaluated as positive by the CIE L*a*b* method. The essence of the evaluation of the lectin histochemistry results is to find a fragment of the target substance, i.e., carrageenan. In the case of computer-assisted analysis, evaluation is more difficult, because the b* value may be skewed by non-specific binding of lectin to the matrix. For products with undeclared addition of carrageenan and a positive result from Human Inspection, the value of the b* parameter is partially intertwined with the negative and positive samples as seen in Figure 4 (samples exceeding the LoD). Sample 68-19 was judged to be negative on the basis of the b* threshold. Samples 67-19 and 153-19 exhibited a large variability in the b* value between measurements (Table 2 and Figure 4). This dispersion could have been caused by an uneven distribution of carrageenan in the products, or a non-specific LH reaction

with one of the additives. Even in this case, with regard to the large dispersion, it can be assumed that the substance was unevenly distributed. Samples 68-19, 70-19, 146-19, 147-19, 148-19, and 154-19 confirmed the results of the Human Inspection, and were to be considered positive on the basis of the CIE L*a*b* method.

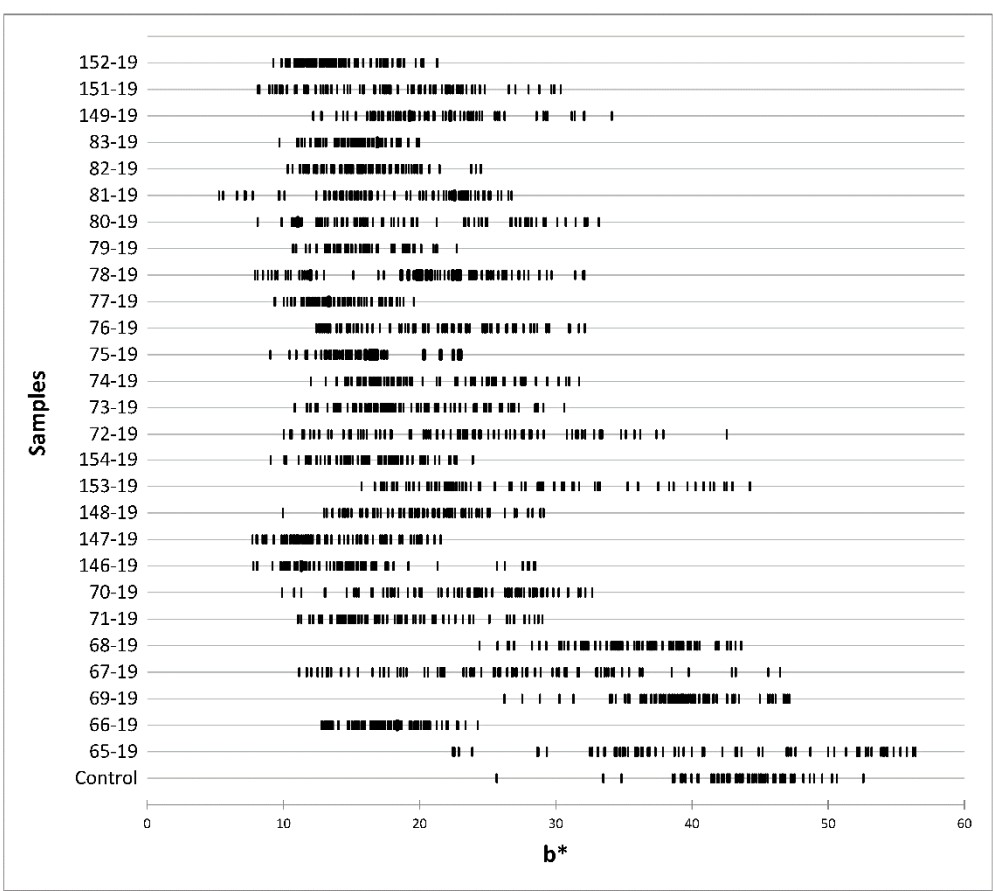

**Figure 4.** Individual value plot of CIE b* measurement variability of retail market samples.

Samples declared positive were also confirmed to be positive by the CIE L* a* b* method. The largest variability was confirmed in sample 72-19 (Table 3, Figure 4). However, for this sample, after adding the required measurement uncertainty (3 × SD), a positive measurement result could not be guaranteed. As with false positive samples, uneven distribution of carrageenan in the cheese matrix can be assumed in this case as well. This may be due to the low concentration of carrageenan used or the technological processing of adding carrageenan to the product. Usually, carrageenan is added to foodstuffs in an amount of 0.005–2.0% by weight [6].

Statistically significant differences were also demonstrated between individual products. Their reason was not verified in the work, we assume that it was mainly a different concentration of the carrageenan used, as well as different types of carrageenan. With regard to applicable legal provisions, the manufacturer is not obliged to provide this information. The only difference in labeling is between refined and semi-refined carrageenan. Refined carrageenan is designated as E407, while semi-refined is designated as E407a [57]. The differences shown are given in Table 3 and point to the possibility of further discrimination by the CIE L*a*b* method for processed cheese.

Validation of the CIE b* method on marketable products confirms that the detection of carrageenan in the cheese matrix can also be performed by changing the color of the matrix in the b* value. In addition, not only due to the presence of characteristically formed carrageenan with a positive LH reaction (Figures 3 and 5), but also due to a change in the color of the protein matrix (Figure 3), we assume that the color change of the cheese

matrix (b*) occurs due to the specific binding of DAB to carrageenan galactose that is bound to the protein network. Carrageenan forms a strong bond with protein [58]. A false negative reaction can also occur with non-specific protein–protein binding, when the carbohydrate is insufficiently recognized by lectin [59]. Even with related methods, a false positive reaction may occur, for example, in the case of immunofluorescence, where a false positive reaction occurs due to autofluorescence or non-specific binding of antibodies [60]. Immunohistochemistry can also give false positive results due to specific protein binding [61].

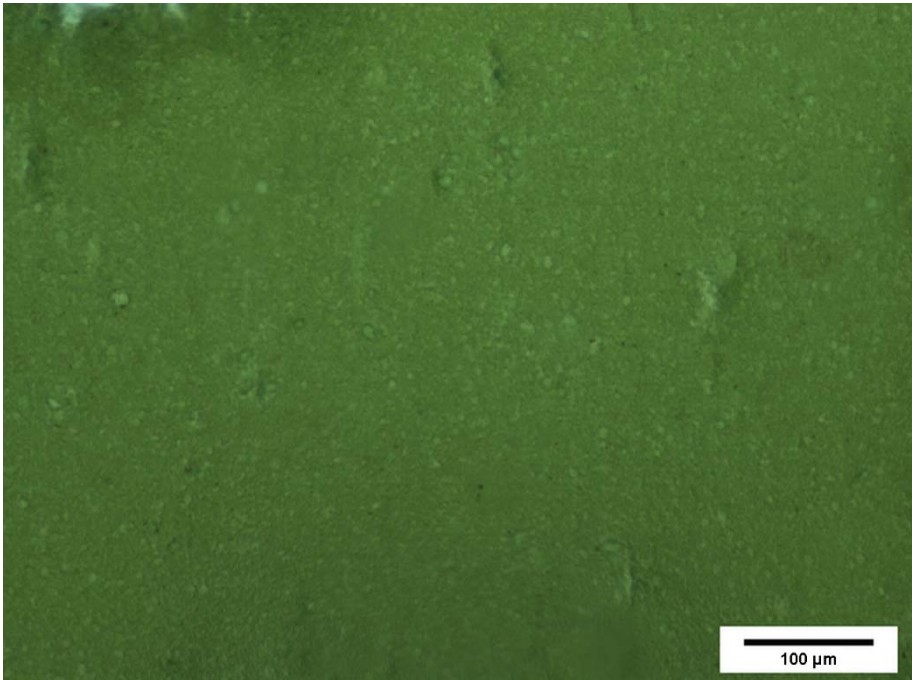

**Figure 5.** Negative sample of processed cheese, lectin histochemistry, background staining B-Calleja (green—protein).

## 4. Conclusions

Fixation of soft matrices in histology is very problematic. In this work, AFA solution was confirmed as the most suitable fixation medium. This fixation medium achieves the optimum hardness of the processed cheese for subsequent histological processing. It was found that carrageenan could be detected in processed cheese by lectin histochemistry. Concentration of 2 $\mu$g mL$^{-1}$ of *Arachis hypogaea* lectin was confirmed for optimal color intensity of the positive reaction. The Limit of Detection (LoD) was established at 100 mg kg$^{-1}$ for Human Inspection and 43.64 $\pm$ 12.33 for CIE L*a*b*. The methods were also validated on samples from the retail market. The results of Human Inspection agree in most cases with the declaration of carrageenan on the packaging, except for nine samples, which were declared negative by the manufacturer and in six samples for which the result was confirmed by computer-assisted analysis. For three samples, the confirmation was not conclusive within the LoD. The CIE L*a*b* results indicate that the computer-assisted analysis may be a suitable complementary analysis to Human Inspection. The most suitable parameter for measurements in the CIE L*a*b* color space was the b* parameter. In the absence of a reference method for the detection of carrageenan in food, further research should compare the available methods for its detection.

**Author Contributions:** The authors' responsibilities were as follows: Conceptualization: M.P. and M.B.; Methodology, M.B.; data curation, M.P.; writing—original draft preparation: M.B. and Z.J. writing—review & editing: Z.J., M.B. and M.P.; visualization, Z.J.; supervision: B.T. and M.B.; project

administration: B.T.; funding acquisition: B.T. All authors have read and agreed to the published version of the manuscript.

**Funding:** This research was funded UVPS Brno project No. FVHE/Tremlová/ITA2019.

**Institutional Review Board Statement:** Not applicable.

**Informed Consent Statement:** Not applicable.

**Conflicts of Interest:** No potential conflict of interest was reported by the authors.

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
