# Peer review of "Detection of Carrageenan in Cheese Using Lectin Histochemistry"

_applsci, doi:10.3390/app11156903_

Round 1
Reviewer 1 Report
The paper submitted by Marie Bartlova and collaborators describes that lectin histochemistry can be used to detect carrageenan in processed cheese with human inspection and computer-assisted analysis. The approach to use lectin histochemistry for food analysis is interesting. The data represented in this paper are reliable. The reviewer has some specific comments as below:
- In Introduction, the author should mention about the significance to detect carrageenan in processed cheese.
- In L251, although the author mentions as “Products with undeclared addition of carrageenan was confirmed by the CIE L*a*b* method as well”, the reviewer do not think so. The carrageenan detecting method using lectin histochemistry is not an established method: THIS paper is the one to try to establish. To confirm the results of the human inspection and computer-assisted analysis, analyzing with another method(s) such as colorimetric methods, potentiometry, chromatography or microscopic methods as described in L224-233 by the author is essential. The reviewer considers the possibility of adulteration of other additives which contain terminal Gal in its molecule.
- Figure 3, the reviewer can not understand what the figure represents. At least, the titles of axes are too small to see.
The followings are minor comments:
- In L31, although the author mentions as “The strength of the gel depends on the carrageenan-casein binding”, not all of processed cheese contain carrageenan. This is limited in carrageenan-added processed cheese.
- In the section 2.3. Lectin Histochemistry (L96), the author should represent the lectin information which is used for LH with reference(s) which shows the sugar binding specificity of the lectin, especially binding specificity to carrageenans.
- In Figure 2, a title of y-axis with unit is required.
- In L173 and L177: Arachis Hypogeae lectin --> Arachis hypogeae lectin
- In the legends of Figures 4 and 5, the author should explain about “B-Calleja”.
Reviewer 2 Report
This manuscript presents basic scientific flaws that influences negatively the interesting work carried out by the authors. In my opinion, this work can be only accepted for publication after an important revision by the authors. In the following my suggestion and comments:
1) Abstract has to stand the work by itself. No justification for carrageenan detection, just mentioned in Lines 48-49 (introduction). Please add. Improve sentence in Lines 17-19, for instance, the word suitable was mentioned twice.
2) Lines 70-72: Dimensions of the cheese samples during TPA measurements? Please add.
3) Statistical processing: Please add significance (0.05 or 0.01, or both?)
4) Serious flaw: Please edit your ALL graphs with better resolution and improve their quality as a Q1 journal requires!!. UNITS, BARS, AXIS, CAPTIONS, TREATMENTS..... The same for FIGURE 3.
5) Please revise the statistical analysis applied in Tables 1 and 3, NOT sure about the differences reported.
6) As a suggestion, try to find a nicer way to present the results reported in Table 4, statistics in this way are just confusing the reader.
Round 2
Reviewer 1 Report
Although the answers by the author almost made sense to the reviewer, the reviewer still has several comments as follows:
L271-L281:
The author lacks a perspective on the possibility that LH using PNA detects components having a terminal b1-3 galactose residue other than carrageenan. To confirm the result of LH, the author should check the carrageenan presence in the LH-positive sample which was negative according to the manufacturer by other methods as the reviewer previously suggested.
Figure 3:
The reviewer still cannot understand the Figure 3. Why does the author use the scatter plot? What is the meaning of “order of measurement”? Does the order effect to the result significantly? Scatter plot is generally used to represent the relationship between two different numeric variables. “Order of measurement” is just an order, not an variable. The reviewer recommends to make a graph with individual value plots instead.
L106:
The sentence in L106-L108 is not appropriate. In fact, PNA binds to β-D-Gal (1 → 3) -D-GalNAc, but carrageenan does not contain this structure, especially D-GalNAc.
And the original paper that PNA recognizes carrageenan should be shown as a reference.
Table 4:
Sample Nos. and the numbers of positive selections/negative selections are very confused. The reviewer recommend to change the notation of sample Nos.: for an example, 65/19 --> 65-19.
Others
L104:
Biotinylated lectin was used to detect carrageenan of Arachis hypogaea (Sigma-Aldrich, USA) at a concentration of 2 μg ml-1.
--> Biotinylated Arachis hypogaea lectin (peanut agglutinin, PNA)(Sigma-Aldrich, USA) at a concentration of 2 μg ml-1 was used to detect carrageenan.
L118:
Computer-Assisted Analysis of CIE L*a*b*
→• Computer-Assisted Analysis of CIE L*a*b*
L279:
The figure contains brown fragments of possible carrageenan, although this sample should have been negative according to the manufacturer.
L282:
…was confirmed… --> ... was examined …
(“Confirm” is not appropriate here.)
Reviewer 2 Report
Changes have been made accordingly. The article can be published in present form.
Author Response
Dear reviewer,
Thank you for your comments and suggestions that allowed us to greatly improve the quality of the manuscript.